Validation and clinical application of a targeted next-generation sequencing gene panel for solid and hematologic malignancies

Prieto-Potin Iván 1
Carvajal Nerea 1
Plaza-Sánchez Jenifer 1
Manso Rebeca 1
Aúz-Alexandre Carmen Laura 1
Chamizo Cristina 1
Zazo Sandra 1
López-Sánchez Almudena 1
Rodríguez-Pinilla Socorro María 1
Camacho Laura 2
Longarón Raquel 2
Bellosillo Beatriz 2
Somoza Rosa 3
Hernández-Losa Javier 3
Fernández-Soria Víctor Manuel 4
Ramos-Ruiz Ricardo 4
Cristóbal Ion 5
García-Foncillas Jesús 5
Rojo Federico frojo@fjd.es 1
1 Department of Pathology, CIBERONC, UAM, Fundación Jiménez Díaz University Hospital Health Research Institute , Madrid , Spain
2 Department of Pathology, Hospital Del Mar Medical Research Institute , Barcelona , Spain
3 Department of Pathology, Vall d’Hebron University Hospital , Barcelona , Spain
4 Genomics Unit, Madrid Science Park , Cantoblanco, Madrid , Spain
5 Translational Oncology Division, UAM, Fundación Jiménez Díaz University Hospital Health Research Institute , Madrid , Spain
Evans D. Gareth
Electronic publication date: 2020 Oct 6
Publication date: 2020
Volume: 8
Electronic Location ID: e10069
Received 2020 Jul 2; Accepted 2020 Sep 9
Copyright: ©2020 Prieto-Potin et al.
Copyright year: 2020
Copyright holder: Prieto-Potin et al.
License: This is an open access article distributed under the terms of the Creative Commons Attribution License, which permits unrestricted use, distribution, reproduction and adaptation in any medium and for any purpose provided that it is properly attributed. For attribution, the original author(s), title, publication source (PeerJ) and either DOI or URL of the article must be cited.
License URL: https://creativecommons.org/licenses/by/4.0/

Keywords: Next-generation sequencing, Validation, Cancer, Solid tumor, Hematological malignancies

Funding: Spanish Ministry of Economy and Competitiveness (MINECO, AES program) PI15/00934 PI18/00382 Ministry of Health (Spanish Biomedical Research Center in Cancer, CIBERONC) CB16/12/0241 Fundación Jiménez Díaz Biobank PT13/0010/0012 Institute of Health Carlos III (ISCIII/FEDER) intensification program, ProteoRed PRB2-ISCIII PT13/0001 Work at Fundación Jiménez Díaz University Hospital was supported by grants from the Spanish Ministry of Economy and Competitiveness (MINECO, AES program, PI15/00934, PI18/00382) and the Ministry of Health (Spanish Biomedical Research Center in Cancer, CIBERONC CB16/12/0241). Iván Prieto-Potin is supported by the Fundación Jiménez Díaz Biobank grant PT13/0010/0012 and Federico Rojo is the recipient of grants from the Institute of Health Carlos III (ISCIII/FEDER) intensification program, ProteoRed PRB2-ISCIII PT13/0001. The funders had no role in study design, data collection and analysis, decision to publish, or preparation of the manuscript.

==============================
Background

Next-generation sequencing (NGS) is a high-throughput technology that has become widely integrated in molecular diagnostics laboratories. Among the large diversity of NGS-based panels, the Trusight Tumor 26 (TsT26) enables the detection of low-frequency variants across 26 genes using the MiSeq platform.

Methods

We describe the inter-laboratory validation and subsequent clinical application of the panel in 399 patients presenting a range of tumor types, including gastrointestinal (GI, 29%), hematologic (18%), lung (13%), gynecological and breast (8% each), among others.

Results

The panel is highly accurate with a test sensitivity of 92%, and demonstrated high specificity and positive predictive values (95% and 96%, respectively). Sequencing testing was successful in two-thirds of patients, while the remaining third failed due to unsuccessful quality-control filtering. Most detected variants were observed in the TP53 (28%), KRAS (16%), APC (10%) and PIK3CA (8%) genes. Overall, 372 variants were identified, primarily distributed as missense (81%), stop gain (9%) and frameshift (7%) altered sequences and mostly reported as pathogenic (78%) and variants of uncertain significance (19%). Only 14% of patients received targeted treatment based on the variant determined by the panel. The variants most frequently observed in GI and lung tumors were: KRAS c.35G > A (p.G12D), c.35G > T (p.G12V) and c.34G > T (p.G12C).

Conclusions

Prior panel validation allowed its use in the laboratory daily practice by providing several relevant and potentially targetable variants across multiple tumors. However, this study is limited by high sample inadequacy rate, raising doubts as to continuity in the clinical setting.

Introduction

Most molecular diagnostics laboratories have incorporated next-generation sequencing (NGS) technology, which allows high-throughput sequencing of the genome. This machinery has been introduced in part to meet the need of clinicians to gather data on genetic alterations and precisely guide decisions on specific molecular-based therapy approaches for individual patients (Friedman et al., 2015). The substantial cost-effectiveness and multiple advantages of NGS over other technologies may explain its widespread use (Tan et al., 2018). One single assay of NGS testing allows simultaneous screening of multiple genes in numerous samples in comparison to the rest of diagnostic platforms that generally analyze an individual gene of a unique sample. NGS is a highly sensitive tool requiring small amounts of DNA input to provide variant allele frequencies (Luthra et al., 2015; Surrey et al., 2016).

Molecular profiling of tumoral DNA is a key attribute of massive sequencing that permits the identification of only a few driver or resistance mutations required for the administration of accurate treatments (Morganti et al., 2019). Hence, platforms analyzing a single marker are becoming obsolete as laboratory teams opt for techniques that yield results from several markers at once. An NGS-based gene panel test can be used to detect genetic aberrations in different biomarkers that can be targeted by molecular-based drugs (Nagahashi et al., 2018). Accordingly, the development of distinct gene panels enables identification of multiple mutations of a particular tumor type. In lung cancer, alterations in the EGFR, ALK or ROS1 genes are used to guide FDA-approved therapies (Hyman, Taylor & Baselga, 2017). Among the many NGS-based panel types, the Trusight Tumor 26 (TsT26) by Illumina uses a small actionable gene panel facilitating the identification of low-frequency variants of genes involved in targeted therapy for solid tumors (Dong et al., 2015). The panel included the KRAS, NRAS and BRAF genes that may be used to determine the eligibility of colorectal cancer (CRC) patients for targeted anti-EGFR treatment, as well as to establish prognosis at any stage of the disease (Sepulveda et al., 2017). Several guidelines and recommendations have been published to standardize the implementation of NGS-based panels in the clinical setting by means of prior technical validation (Jennings et al., 2017). An NGS-based panel should not be set up in a clinical practice unless an acceptable validation is performed beforehand (Matthijs et al., 2016). In fact, the validation process should thoroughly document how the assay is reliable in identifying known mutations detected by diagnostic standards (McCourt et al., 2013).

After multidisciplinary clinical consensus was reached, it became clear that an NGS-based panel would be needed in routine medical care for detailed molecular characterization of patients presenting a range of advanced cancer types. A unique molecular testing could satisfy the demand by considering either administering targeted therapy or selecting appropriate candidates to participate in early-stage clinical trials from our institution. We postulated that a small gene panel such as the TsT26 panel including several genes implicated in targeted therapy and targets required for the recruitment to specific early-stage clinical trials would be suitable to fulfill our care necessity. In order to test the panel capacities, the study first aimed to demonstrate whether it could be used to determine the mutational status of three precise genes (i.e., KRAS, NRAS and BRAF), all of which are associated with treatment decision-making in CRC. For that purpose, we managed an inter-laboratory validation to incorporate the assay in routine clinical practice. Between 2015 and 2017, the laboratory subsequently employed the panel in usual activity. Second, we describe successful use of the TsT26 panel in 399 patients presenting diverse tumorigenesis and evaluate the utility of the panel in the clinical context.

Material and Methods

FFPE tissue collection

The TsT26 panel performance was conducted in three clinical centers: Hospital Del Mar Medical Research Institute (Barcelona, Spain, n = 16), Vall d’Hebron University hospital (Barcelona, Spain, n = 16) and Fundación Jiménez Díaz University hospital (Madrid, Spain, n = 16). The Madrid Science Park was an additional collaborating entity. It overall included archived formalin-fixed, paraffin-embedded (FFPE) tissue material from 48 patients presenting primary colorectal cancer with a previously determined mutational status of KRAS, NRAS and BRAF genes. Samples were obtained from MARBiobank (PT17/0015/0002), VHIR-Biobank (PT17/0015/0026) and the Biobank Fundación Jiménez Díaz (PT17/0015/0006), each belonging to the Spanish National Biobanks Network. Additional FFPE samples from a variety of tumor types were used in TsT26 panel testing at the Fundación Jiménez Díaz University hospital (Madrid, Spain, n = 399). Written consent was received from each donor . All investigations followed standard operating procedures with the approval of the Fundación Jiménez Díaz University hospital Ethics and Scientific Committee (PIC 23-2012) and were conducted in accordance to the principles outlined in the Declaration of Helsinki.

Quality control (QC) 1 for tumor-cell content

FFPE tissue sections (4 µm thick) were obtained for hematoxylin and eosin staining (Dako Coverstainer, Agilent, Santa Clara, CA, USA) to ensure tumor-cell content (TCC) of at least 30%. A pathologist (SMR-P and FR) examined the samples for tumor-cell content, scored them for percentage of neoplastic nuclei and circled the tumor area. For TCC of below 30%, macrodissection was performed with a scalpel blade (Fig. S1A).

DNA isolation

Consecutive FFPE tissue sections were obtained to extract genomic DNA according to the specimen type. Surgical resections were sectioned 10 µm thick, biopsies 30 to 40 µm, and endoscopies and cytologies 100 µm deep. Isolation was done by using the cobas® DNA Sample Preparation Kit (Roche Diagnostics, Pleasanton, CA, USA) according to the manufacturer’s protocol. Both concentration and purity were determined by Nanodrop spectrophotometer (Thermo Fischer, Waltham, MA, USA) and Qubit 3.0 fluorometer (Thermo Fisher, Waltham, MA, USA).

TsT26 ΔCq DNA QC2

Extracted DNA was amplified in triplicate by quantitative PCR using the KAPA SYBR FAST master mix (Life technologies, Grand Island, NY) on the Lightcycler® 480 system (Roche Molecular System, Pleasanton, CA, USA). The amount of DNA input was established by comparing the ability of DNA to be amplified in relation to a non-FFPE reference genomic DNA. A ΔCq value was calculated for each sample as follows: ΔCq= mean sample Cq value - mean non-FFPE control Cq value. A mean of ΔCq<6 was considered appropriate for library preparation despite the instructions of the protocol, which recommended ΔCq<4 (Fig. S1B).

TsT26 Library preparation QC3

NGS libraries were prepared using the TsT26 panel (Illumina, San Diego, CA, USA) (Table S1), a multiplexing kit of 178 amplicons covering 82 exonic regions across 26 genes (Table 1), as indicated by the TsT26 reference guide. The obtained products were checked for their base pair range using 2% agarose gel electrophoresis along with a 50 bp ladder (Sigma-Aldrich, San Luis, USA) or a 2100 bioanalyzer instrument (Agilent, Santa Clara, CA, USA) (Fig. S1C). Generated libraries in the 300-330 base pair range were considered suitable for sequencing. Library concentration was measured using the Qubit 3.0 fluorometer (Thermo Fisher, Waltham, MA, USA) and normalized to 4 nM in elution buffer with Tris.

Table 1 Trusight®Tumor 26 assay exon coverage by amplicons (82 exons from 26 genes were covered by 178 amplicons).

Gene symbol	Accession number	Exons covered	Number of amplicons to exon coverage	
AKT1	NG_012188.1	2	1	
ALK	NG_009445.1	23	1	
APC	NG_008481.4	15a	14	
BRAF	NG_007873.3	11,15	3	
CDH1	NG_008021.1	8,9,12	6	
CTNNB1	NG_013302.2	2	2	
EGFR	NG_007726.3	18,19,20,21	7	
ERBB2	NG_007503.1	20	2	
FBXW7	NG_029466.2	7,8,9,10,11	13	
FGFR2	NG_012449.2	6	2	
FOXL2	NG_012454.1	1	1	
GNAQ	NG_027904.2	4,5,6	6	
GNAS	NG_016194.2	6,8	2	
KIT	NG_007456.1	9,11,13,17,18	9	
KRAS	NG_007524.1	1,2,3,4	8	
MAP2K1	NG_008305.1	2	1	
MET	NG_008996.1	1,4,13,15,16,17,18,20	22	
MSH6	NG_007111.1	5	3	
NRAS	NG_007572.1	1,2,3,4	8	
PDGFRA	NG_009250.1	11,13,17	5	
PIK3CA	NG_012113.2	1,2,7,9,20	15	
PTEN	NG_007466.2	1b,2,3,4,5b,6b,7,9	17	
SMAD4	NG_013013.2	8,11	5	
SRC	NG_023033.1	10	2	
STK11	NG_007460.2	1,4,6,8	7	
TP53	NG_017013.2	2c,3c,4c,5c,6c,7,8c,9c,10,11	16	
Notes.

a exon 15 of the APC gene was split into three regions and each covered respectively by two, two and 10 amplicons.

b exons 1, 5 and 6 of the PTEN gene were split into two regions each and separately covered by two, two and two amplicons.

c exons 2, 3 and 4 of the TP53 gene were together covered by six amplicons; as well as exons 5 and 6, 8 and 9, respectively by four and three amplicons.

TsT26 high-throughput sequencing

Libraries were then diluted to 10 or 12 pM and to 15 or 20 pM and pooled on a v2 300-cycle or v3 600-cycle sequencing kits according to the manufacturer’s protocol. Sequencing was achieved for both pool A and B by loading 600 µl of library mixes. Some runs were loaded along with 1% PhiX.

TsT26 analysis, quality metrics and variant detection

The integrated analysis software (Illumina, San Diego, CA, USA) including image analysis, base calling and assignation of quality scores automatically performed primary analysis. The sequencing analysis viewer software (SAV, Illumina, San Diego, CA, USA) confirmed quality metrics by using interop files along with run info and parameters. A Phred score of Q30 was considered for each run. The MiSeq Reporter software (Illumina, San Diego, CA, USA) included demultiplexing, sequence alignment and variant calling. Successful sequencing runs generated 2 FASTQ files, 2 BAM and BAM-BAI files for each sample pool A and pool B library pair and a single genomic variant call file (VCF). Integrative Genomics Viewer software (IGV, Broad Institute, CA, USA) enabled to visualize sequenced regions (Thorvaldsdóttir, Robinson & Mesirov, 2013). An exportable excel format was generated for amplicon coverage assessment.

Annotation of detected variants was performed using the Illumina Variant Studio version 2.2 software (Illumina, San Diego, CA, USA). Every variant with a variant allele frequency (VAF) less than 3% was filtered and excluded before review. Detected variants were marked with a PASS filter flag if the following criteria were met: variant must be present in both pools, cumulative depth of 1000x or an average depth of 500x per pool. Those detected variants that did not satisfy this criterion or presented strand bias were further assessed during interpretation. A biologist (NC or SZ or CC) evaluated variants by identifying missense, frameshift, stop gain or loss, or in-frame insertion- or deletion- affected sequences. Variant classification employed ClinVar (http://www.ncbi.nlm.nih.gov/clinvar) (Harrison et al., 2016; Landrum et al., 2016), COSMIC (http://cancer.sanger.ac.uk/cosmic) (Tate et al., 2019) and cbioPortal (http://www.cbioportal.org/) (Cerami et al., 2012; Gao et al., 2013) databases. Additional catalogues such as CIVIC (https://civicdb.org/home) (Griffith et al., 2017), OncoKB (https://oncokb.org/) (Chakravarty et al., 2017) or the Cancer Genome Interpreter (https://www.cancergenomeinterpreter.org/analysis) (Tamborero et al., 2018) were also accessed for variant interpretation. Pathogenic, likely pathogenic, variant of uncertain significance (VUS) and benign or likely benign variants were reported according to standard guidelines (Richards et al., 2015; Hoskinson, Dubuc & Mason-Suares, 2017). A pathologist (SMR-P and FR) provided final authentication of the reported variants.

KRAS and NRAS pyrosequencing

Pyrosequencing was determined for the KRAS and NRAS genes using the CE-IVD therascreen KRAS, NRAS and RAS Extension pyro kits (Qiagen, Hilden, Germany), according to the manufacturer’s instructions.

BRAF cobas assay and direct sequencing

The CE-IVD cobas® 4800 BRAF V600 Mutation Test was used to identify BRAF c.1799T>A (p.V600E) mutation by real time PCR technology on the cobas z 480 Analyzer (Roche Diagnostics, Pleasanton, CA, USA), in agreement with the manufacturer’s protocol.

The mutational status of BRAF was also determined by direct sequencing using the ABI-Prism 3730 XL DNA analyzer (Applied Biosystems Foster City, CA, USA), as previously described (Bessa et al., 2008). Primers were designed with Primer Express software (Applied Biosystems, Foster City, CA, USA) using BRAF sequences NG-007873.3: BRAF-Fw: 5′-CTCTTACCTAAACTCTTCATAATGCTTGC-3′ and BRAF-Rv: 5′-CAGCATCTCAGGGCCAAAAA-3′.

Statistical analysis

We hypothesized that the expected difference of the detected variants found by the NGS-panel in comparison with the reference standard should not exceed 10%. By using the PS program (Dupont & Plummer, 1990), the minimal sample needed to detect this difference was set to 44 samples with a power of 0,90 and a two-sided error alpha of 0,05. For sensitivity analysis, test variants were assigned either a true positive (TP) if detected or false negative (FN) if not detected. Sensitivity was calculated as the proportion of samples with a detected variant, TP/(TP+FN). We estimated specificity as the proportion of cases without a detected variant, also considered as TN/(FP+TN), that is to say variants not detected by the standard reference method that the NGS-panel identified as not detected. FP= false positives and TN= true negatives. The accuracy or extent of agreement between the outcome of the two methods was determined as (TP+TN)/(TP+FP+FN+TN). Confidence intervals (CI) were calculated by the method of Clopper and Pearson, as previously described (Van Stralen et al., 2009; Mattocks et al., 2010; Lih et al., 2017). A pairwise comparison using the Mann–Whitney test was applied between the two employed chemistries. A p-value of less than 0,05 was considered significant. Statistical analysis used SPSS version 21.0 software for Windows (IBM, New York, NY, USA) and GraphPad Prism version 5.0 software (GraphPad Software, Inc., La Jolla, CA). Descriptive data were expressed as the mean and 95% CI.

Results

Inter-laboratory performance of the TsT26 panel

To assess the performance of the NGS-based panel, we compared the outcomes of the test with a reference standard to identify variants in the KRAS, NRAS and BRAF genes. Pyrosequencing, RT-PCR and Sanger sequencing constituted the conventional methods employed as reference standard that detected 29 variants out of the 48 selected samples, the other 19 exhibited a wild-type genotype. Performance was calculated as the whole data produced by each center, that is to say 48 samples run in triplicate or 144 outcomes taken into consideration for the agreement analysis. Sensitivity, specificity and accuracy were measured to evaluate the performance of the NGS-panel test in order to describe either detection or not detection of the variants in the mentioned genes, as indicated in Table 2. Regarding the limit of detection, the observed VAFs of the variants detected by the NGS-panel ranged from 3,23% to 68,97%. Sample 1-5 variant KRAS c.35G>C (p.G12A) was not detected and samples 3-4 and 3-11 were identified as variant detected, respectively as KRAS c.34G>A (p.G12S) and c.35G>T (p.G12V) (Table S2). So, the variant calling finally identified thirty variants. Center 1 was able to detect 25 variants whereas centers 2 and 3 both distinguished 29 detected variants (Fig. 1A).

Table 2 TsT26 panel performance by determining the mutational status of KRAS, NRAS and BRAF genes.

TsT26	Gold standard	
	Detected variant	Not detected variant	Total	
Detected variant	80	3	83	
Not detected variant	7	54	61	
Total	87	57	144	
Sensitivity	92% (80/87, 95% CI [84–97])	
Specificity	95% (54/57, 95% CI [85–99])	
Accuracy	93% (134/144, 95% CI [88–97])	
Positive predictive value	96% (80/83, 95% CI [90–99])	
Negative predictive value	88% (54/61, 95% CI [79–94])	

Figure 1 Sequencing quality metrics of the Trusight® Tumor 26 panel during the validation procedure of the mutational status of KRAS, NRAS and BRAF genes.

(A) Variants detected by the reference standard method against the Trusight® Tumor 26 panel. Data are shown as percentage. (B) Cluster density and cluster passing filter quality metrics, respectively expressed in cluster per mm2 and percentage. (C) Read depth of detected variants, 20K represents a depth of 20000x. (D) Variant allele frequency of each gene is shown as a percentage. Data are represented as box and whisker plots with median and IQR.

High-throughput sequencing quality metrics of the panel performance

Nine runs employing v2 300-cycle sequencing chemistry were carried out during the validation process. Quality metrics including cluster density and cluster passing filter were found to be slightly increased in comparison to the manufacturer’s guidance as depicted in Fig. 1B. Every detected variant encountered by NGS testing presented a read depth of higher than 1000x and a variant allele frequency (VAF) greater than 3% (Figs. 1C and 1D).

Patient characteristics and clinical practicability of the TsT26 panel

A set of 399 patients was included in this study. Sex and age data were available for 386 and 365 patients, respectively. Regarding tumor types, the most common consisted of gastrointestinal (GI), hematologic, lung, gynecological, and breast samples, whereas melanoma, head and neck, genitourinary, central nervous system (CNS), and other histological cancer types were limited in number (Table S3). Overall, the sample set consisted of biopsy specimens, surgical resections, endoscopies and cytologies (Table 3). From the entire data set, 40% was from external origin.

Table 3 Clinical and patient characteristics.

Characteristics	Number of patients	Cytology	Resection	Endoscopy	Biopsy	
Sex, no. (%)a						
Female	188(49)					
Male	198(51)					
Mean age, y.o. (95%CI)a	59(58–61)					
Tumor type, no. (%)						
Gastrointestinal	115(29)	0	35(30)	23(20)	57(50)	
Hematologic	73(18)	0	12(16)	0	61(84)	
Lung	51(13)	13(25)	6(12)	0	32(63)	
Gynecological	38(8)	0	14(34)	1(3)	23(63)	
Breast	33(8)	2(6)	9(27)	1(3)	21(64)	
Genitourinary	20(5)	0	9(47)	0	11(53)	
Head and Neck	19(5)	0	6(32)	0	13(68)	
Melanoma	15(4)	1(7)	3(20)	0	11(73)	
Central Nervous System	10(3)	0	3(30)	0	7(70)	
Other solid tumor	25(6)	1(4)	3(12)	0	21(84)	
Notes.

a Sex and age data was not available for every patient included in the study.

Two-thirds of all samples were successful in sequencing testing, while one-third failed due to unsuccessful quality-control filtering (Figs. 2A and 2B). Quality controls included initial TCC management, followed by assessment of DNA quality and final quantitation of library preparation. Failure to meet any of these quality controls lead to sequencing failure (Fig. 2C).

Figure 2 Practicability of the Trusight® Tumor 26 panel.

(A) Study design of the panel performance. (B) Flow diagram depicting the number of FFPE samples that either succeded or failed to NGS testing. (C) Workflow followed by each of the 399 FFPE samples included in TsT26 panel study. Samples underwent diverse quality controls (QC). QC1 referred to the tumor-cell content; a cut off value was established in 30%. Note that samples between 10–30% with no possibility of macrodissection underwent direct DNA isolation. QC2 indicated the quality of the sample in comparison to a fresh commercial preserved sample; a ΔCq value less than 6 was acceptable to continue the library preparation. QC3 determined the fragmentation of the library, library products of less than 300 bp were not considered for sequencing.

DNA quality assessment for high-throughput sequencing by quantitative PCR

Most samples presenting a ΔCq<4 resulted in a successful NGS sequencing. Five samples were sequenced despite of showing ΔCq>6 upon explicit clinical request. Several samples exhibiting a ΔCq value between 4 and 6 were able to generate valuable libraries. Consequently, we extended the cut-off value of the ΔCq to 6. Thirty-seven samples did not undergo DNA quality assessment, 5% of them resulting in failed NGS, 3% in detected variants and 1% in not detected variant (Table 4). Samples that failed sequencing were to the most extent biopsy specimens (26%). Almost half of them had a poor concentration (13%) and were from external origin (13%), as well as exhibited either a low TCC or inadequate representative tumor material (8%).

Table 4 DNA quality assessment by quantitative PCR.

	ΔCt<4	4<ΔCt<6	ΔCt>6	
Detected variant	159 (40%)	19 (5%)	3 (1%)	
Not detected variant	59 (15%)	5 (1%)	2 (1%)	
NGS fail	26 (6%)	27 (7%)	62 (15%)	

Variant detection and high-throughput sequencing quality metrics during TsT26 panel implementation

Detected variants were identified in 74% (194 samples) of the successfully sequenced samples, whereas in 26% (69 samples) variants were either not detected or a wild type genotype was found (Table S4). The highest number of detected variants was observed in the TP53 (28%), KRAS (16%), APC (10%) and PIK3CA (8%) genes. In contrast, a lower amount of detected variants was encountered in MET (5%), BRAF (4%), SMAD4 (3%), as well as in KIT, PTEN, NRAS, CTNNB1, FBXW7, CDH1, HER2, (2% each) and GNAS, MAP2K1, STK11, EGFR, PDGFRA, MSH6, FGFR2, GNAQ, SRC (1% each). No variants were detected in the AKT1, ALK and FOXL2 genes (Fig. 3A).

Figure 3 Sequencing quality metrics of the Trusight® Tumor 26 panel during clinical implementation.

(A) Total detected variants per gene type identified in the 399 samples tested. (B) Cluster density and cluster passing filter quality metrics respectively expressed in cluster per mm2 and percentage. (C) Read depth of detected variants, 20K represents a depth of 20000x. (D) Variant allele frequency of each gene is shown as a percentage. Data are represented as box and whisker plots with median and IQR.

Thirty-seven runs were held during clinical implementation, 17 employing v2 300-cycle and 20 using v3 600-cycle sequencing chemistries. Although certain runs experienced underclustering, the mean cluster density was found within the optimal range recommended by the manufacturer between 1000 and 1400 clusters per mm2 (K/mm2). Accordingly, these runs showed a high percentage value of cluster passing filter that lead to an elevated mean of this parameter (Fig. 3B). We found statistically significant differences between the cluster density and cluster passing filter of the employed v2 and v3 chemistries, 887 (95% CI [735–1,039]) and 1,230 (95% CI [1,074–1,386]) k/mm2, p = 0,0024; and 94% (95% CI [93–96]) compared to 91% (95% CI [88–93]), p = 0,0093, together in accordance with the manufacturer’s guidelines. A read depth of greater than 1000x was observed for each gene, except in two skin melanomas presenting a reduced depth value that subsequently required further corroboration by additional molecular testing. Additionally, every detected variant demonstrated a VAF greater than 3%. The MET gene showed the highest VAF mean in comparison to the other studied genes (Figs. 3C and 3D).

Coverage by amplicon was calculated by obtaining the mean of each amplicon covering each exonic region. The mean coverage of AKT1 exon 2 and STK11 exon 6 did not satisfy the minimum coverage of 1000x required by the panel. In addition, EGFR exon 21, STK11 exons 1, 4, 8 and TP53 exon 11 presented the same condition although this was compensated by cumulatively counting the coverage of the second pool (Fig. S2).

Detected variants analysis by tumor types

In GI tumors, detected variants were more frequently observed in the KRAS (23%), TP53 (22%), APC (16%) and PIK3CA (8%) genes. Furthermore, detected variants were identified in the same genes in gynecological tumors, whereas in lung there were more recurrently perceived variants in only TP53 (33%) and KRAS (21%) genes. In contrast, detected variants in hematologic malignancies were merely seen in the KRAS gene (Fig. 4). In melanoma, 46% of the detected variants were found in BRAF and 15% in NRAS genes. In breast, the TP53 and PIK3CA genes presented a highest number of affected variants in comparison to other genes. Whereas genitourinary, head and neck, and SNC tumor types exhibited at least one detected variant per gene except in TP53, PIK3CA and MET (Fig. S3).

Figure 4 Detected variant frequencies across tumor types.

(A) Gastrointestinal. (B) Hematologic malignancies. (C) Lung. (D) Gynecological. Columns represent samples and rows show genes expressed by the percentage of samples with a detected variant. Detected variants are shown by grey squares whereas more than one detected variant is depicted by black squares.

Three hundred seventy-two detected variants were identified in 23 genes of the 26 identified by the TsT26 panel (no variants were found for AKT1, ALK and FOXL2). The dominant type of detected variants consisted of missense altered sequences (81%); followed by stop gain (9%) and frameshift (7%) affected sequences. Minor alterations corresponded to in-frame deletions (2%), splice region variants (1%), in-frame insertions (1%) and start lost (1%). Most detected variants were reported as pathogenic (78%) or likely pathogenic (1%), whereas 19% of variants were classified as VUS and 2% as benign or likely benign. The variants more frequently observed were: KRAS c.35G>A (p.G12D) and c.35G>T (p.G12V), but also MET c.504G>T (p.E168D) in GI tumors; TP53 c.742C>T (p.R248W) in hematologic malignancies; KRAS c.34G>T (p.G12C) and c.35G>T (p.G12V) in lung; PIK3CA c.3140A>G (p.H1047R) and c.3140A>T (p.H1047L) in breast and BRAF c.1799T>A (p.V600E) in both melanoma and head and neck cancers (Table S5).

Gene mutation frequencies by histological tumor types

We further compared mutation frequencies of the genes presenting detected variants against those encountered in the TGCA database for each histologic tumor type. Similarity was observed when relating mutations frequencies of genes contained in the TsT26 panel to those from the TCGA set for most of the tumors included in the study. For instance, KRAS, TP53, APC, PIK3CA, SMAD4, FBXW7 and BRAF genes presented higher mutation frequencies in CRC. The prevalence of additional genes in other tumor types can be seen in Figs. S4 and S5. Hepatocarcinoma, cervical and mesothelioma tumor types did not present a sufficient number of cases to perform comparisons.

Clinical decision-making subsequent to high-throughput sequencing

After reviewing the medical records of the 194 patients presenting detected variants in a range of genes after applying the TsT26 panel, we were able to associate a subsequent clinical action to a reported detected molecular alteration (Table S6). Arising from the 372 detected variants found, 37% were considered clinically relevant and a treatment decision was attempted in 13%. Only 14% of patients received targeted therapy based on the variant detected by the TsT26 panel (Table S7).

Discussion

In this study, we conducted an inter-laboratory validation of the TsT26 panel based on the detection of alterations in three genes of potential therapy interest. We obtained robust data regarding the detection of relevant and likely targetable variants across multiple tumors from 399 patients, despite a large number of samples that failed strict quality assessments. Reporting of detected variants was supported by adequate sequencing metrics and subsequent clinical decision-making when indicated.

Concerning the discordant results obtained during the validation process, any of the involved centers was able to authenticate the original mutation c.35G>C (p.G12A) reported by center 1 in KRAS. After performing re-test in both previous sections and a second isolation of DNA, the sample still resulted mutated, leading to tumor heterogeneity as a concluding interpretation of the case. On the other hand, the sequencing analysis found the mutational status of samples 3-4 and 3-11 as mutated for KRAS. The two variants exhibited a low VAF (3,23% for c.34G>A (p.G12S) in center 2; 7,71% and 8,37% for c.35G>T (p.G12V) in centers 2 and 3, respectively) that could not be originally detected by the gold standard method, pyrosequencing. Finally, center 3 corroborated the two variants identified by the TsT26 panel using another platform with a more accurate sensitivity employing qPCR technology. Unfortunately, center 1 was unable to recognize any of the two variants during the high-throughput sequencing analysis. Overall, 29 + 29 + 29 = 87 variants represented the 100% of variants detected by the reference standard whereas the NGS-panel was only able to identify 25 + 29 + 29 = 83 variants. This represented a difference of 5% with respect to the total number of variants recognized by the reference standard. Therefore, not exceeding the expected difference to be found in the comparison evaluation. Only center 1 failed in the performance assessment because it only detected 25 variants out of the 29 detected variants to be distinguished. We should also acknowledge certain limitation in our evaluation process as we did not calculate reproducibility after establishing test sensitivity, specificity and accuracy with respect to the reference standard.

The TsT26 panel incorporated rigorous pre-analytical requirements to obtain a favorable sequencing outcome. Samples were primarily evaluated for their TCC. Although a minimum of 30% of TCC was established as the cut-off value, a large number of samples with a lower TCC were selected to begin the panel testing, in part, because the standard reference of the laboratory was established as a 10% value as the minimum TCC, however most of these samples underwent tissue macrodissection. Secondly, DNA quality was assessed by qRT-PCR. Indeed, this estimation is considered a better indicator of amplifiable material than other common methods such as fluorometric and spectrophotometric evaluations, the latter usually overestimating the amount of double stranded DNA (Deans et al., 2017). A final quality control measured library adequacy for sequencing. Despite showing good DNA quality as demonstrated by a favorable qRT-PCR assessment, several samples failed the library-generation procedure as they did not meet the appropriate right base pair size. Therefore, such a strict quality control undertaken before initiating the sequencing may justify the elevated inadequacy rate of samples that failed the TsT26 testing.

Our data reveal a similar prevalence of detected variants in GI, lung and melanoma tumors with previous published NGS results. In 52 colorectal tumors, KRAS, TP53 and APC were the genes affected with most detected variants using the same panel TsT26 (Giardina et al., 2018). The same genes were also more frequently altered in the GI tumors of the study. In another 45 lung adenocarcinomas, TP53, KRAS and PIK3CA showed the highest percentages of detected variants per gene using the ion torrent AmpliSeq Cancer Hotspot v2 assay (Tsongalis et al., 2014). Similarly, TP53, KRAS and APC presented more somatic alterations in the set of lung samples, whereas PIK3CA was represented in a much lower proportion. Others confirmed the elevated quantity of mutations in the KRAS gene despite the use of a limited sample size (Patel et al., 2017) or a much larger data set (Legras et al., 2018). An implementation study employing a customized Ampliseq NGS panel including 35 genes reported BRAF, TERT and NRAS as the most prevalent mutated genes in a set of 100 primary melanoma samples (De Unamuno Bustos et al., 2017). BRAF and NRAS were also the more frequently mutated genes in melanoma samples as described by others (Fisher et al., 2016; Giardina et al., 2018), except the TERT gene which was not included in the NGS panel.

Numerous studies have characterized a prior validation to implement an NGS-based panel commonly employed in the assessment of targeted therapies for solid tumors (Tsongalis et al., 2014; Csernak et al., 2017; Kou et al., 2017; Luthra et al., 2017; Lee et al., 2018; Maxwell et al., 2018; Sussman et al., 2018; Williams et al., 2018). Similarly, other authors corroborated its use on pediatric hematologic malignancies (Kluk et al., 2016) or myeloid neoplasms (Maes et al., 2017). In contrast, another investigation has directly focused on a concrete cancer type such as in non-small cell lung cancer (NSCLC) (Legras et al., 2018). Although several studies validated NGS-based panels on both solid tumors and hematologic malignancies (Cottrell et al., 2014; Garcia et al., 2017), few reports have aimed to demonstrate that the TsT26 panel is a validated method to implement in the clinical practice in a considerable number of varied tumor tissues (Fisher et al., 2016; Giardina et al., 2018). In addition, this panel has also been used to validate another molecular testing platform in 90 NSCLC tumor samples (Quinn et al., 2015). A greater number of validation studies in high-throughput sequencing with its consequent application in the real world would allow a better knowledge about the identification of variants with a clinically relevant significance, thus improving the integration of this technology in the clinical setting. Although the TsT26 panel was indicated for the analysis of solid tumors, we also underwent extra solid tumors types and hematologic malignancies samples across the panel. This is certainly not the most appropriate panel to test hematologic malignancies. Particular customized panels are exclusively designed for that purpose such as the personalized panel including 48 genes in T-cell lymphomas (Manso et al., 2018). Despite that, detected variants were found in the TP53 gene allowing a concrete clinical decision-making and prognosis of several subgroups of lymphomas (Xu-Monette et al., 2012).

Other limitations may be recognized in the present study. Even though the validation analysis exhibited good concordance regarding the KRAS, NRAS and BRAF genes, supplementary verification considering the rest of the targeted genes would probably demand additional authentication. Another constraint concerns the kind of genetic aberration that the panel is able to recognize. Essentially, the panel merely detects either single or multiple nucleotide variants in a restricted number of genes and cannot identify gene fusions. Moreover, concrete exonic regions of the genes AKT1, STK11, EGFR and TP53 were not adequately covered, thus slightly reducing the sum of gene regions analyzed by the panel.

Despite incorporating a limited number of genes, most of the genes included were tightly linked to potential FDA-approved clinical actionability, such as BRAF mutations c.1799T>A (p.V600E) in melanoma regarding dabrafenib, trametinib or vemurafenib treatments, and EGFR tyrosine-kinase domain mutations in NSCLC for afatinib, erlotinib, gefitinib and osimertinib therapies (Hovelson et al., 2015; Paasinen-Sohns et al., 2017). Likewise, PIK3CA mutations in patients with hormone receptor positive and ERBB2-negative advanced breast cancer who previously received endocrine therapy for alpelisib-fulvestrant (André et al., 2019). As well, KIT mutations in GIST for regorafenib, sunitinib and imatinib drugs (Demetri et al., 2013). Novel NGS-targeted panels may include a much larger number of genes to be tested as demonstrated by a panel targeting 170 genes that proved to bring relevant clinical information on diffuse gliomas by improving both diagnosis and prognosis (Na et al., 2019). Other panels can detect further alterations such as fusions and copy number variations in combination with point mutations in an elevated number of genes (Luthra et al., 2017). In fact, the eligibility of the NGS-based panel remains of vital importance according to the kind of alterations that need to be targeted.

Conclusions

Inter-laboratory validation permitted effective NGS-based panel testing in 399 samples of diverse tumorigenesis. Two-thirds of the samples were able to be sequenced and 372 variants were identified. Reporting of clinically relevant variants allowed subsequent clinical decision-making and targeted treatment administration. However, the findings of the study are limited by high sample inadequacy rate (one-third), mainly explained by the strict quality assessments recommended by the manufacturer, thus generating some uncertainty regarding its continuity in the clinical setting.

Supplemental Information

Supplemental Information 1 Quality controls application in the Trusight® Tumor 26 panel sequencing workflow

(A) Selection of the FFPE blocks were made and 2 micron thick sections were stained with haematoxylin and eosin. A pathologist assessed the tumor cell content (TTC) and actions were completed in accordance with the % of the TCC. (B) ΔCq values. Extracted DNA was amplified in triplicate by quantitative PCR. The amount of DNA input was established by comparing the ability of DNA to be amplified in relation to a non-FFPE reference genomic DNA. A ΔCq value was calculated for each sample as follows: ΔCq = mean sample Cq value - mean non-FFPE control Cq value. A mean of ΔCq¡4was considered as appropriate for library preparation, even though values comprised between 4 < ΔCq < 6 were also suitable for library preparation. ΔCq¿6 were discarded. (D) Library quality control. Library obtained products were checked for their base pair range using a 2100 bioanalyzer instrument (Agilent, Santa Clara, CA, USA) or run in 2% agarose gel. Generated libraries in the 300-330 base pair range were considered appropriate for sequencing.

Click here for additional data file.

Supplemental Information 2 Amplicon mean coverage

(A) pool A and (B) pool B libraries. Vertical dashed lines represent a depth of 1000x, 10000x and 20000x, respectively. Data are expressed as the mean and 95% CI.

Click here for additional data file.

Supplemental Information 3 Distribution of detected variants in additional studied tumor types

(A) Breast, (B) Genitourinary, (C) Head and neck, (D) Melanoma, (E) Central nervous system and (F) Other solid tumor. Each column denotes an individual tumor and each row represents a gene. Detected variants are shown by grey squares whereas more than one detected variant of the same gene is depicted in black squares.

Click here for additional data file.

Supplemental Information 4 Distribution of frequently mutated genes resulted from the TsT26 sequencing in comparison to TCGA mutation frequencies across tumor and histological types

(A) Gastrointestinal (GI), n = 86 vs. n = 1737 TCGA-COAD+READ+STCA+ESCA+PAAD+CHOL+LIHC. (B) Colorectal (CRC) adenocarcinoma, n = 45 vs. n = 534TCGA − COAD + READ.(C)Gastroesophagiccarcinoma, n=19 vs.n=619 TCGA − STCA + ESCA.(D)Pancreaticcarcinoma, n=13 vs.n=171 TCGA-PAAD. (E) Biliary duct, n = 7 vs. n=50 TCGA − CHOL.(F)Hematologic, n = 49 vs. n = 173 TCGA-DLBC+LAML. (G) Lymphoma, n = 25 vs. n = 37 TCGA-DLBC. (H) Leukemia, n = 10 vs. n = 136 TCGA-LAML. (I) Lung, n = 31 vs. n = 1049 TCGA-LUAD+LUSC. (J) Gynecologic, n = 23 vs. n = 1311 TCGA-OV+CESC+UCEC+UCS. (K) Ovarian, n = 13 vs. n = 436 TCGA-OV. (L) Endometrial, n =5 vs. n = 529 TCGA-CESC. Data is expressed as percentages values of samples affected with detected variants. Black bars represent mutation frequencies resulted from the TsT26 sequencing whereas white bars correspond to values obtained from the TCGA dataset.

Click here for additional data file.

Supplemental Information 5 Distribution of frequently mutated genes resulted from the TsT26 sequencing in comparison to TCGA mutation frequencies in other cancer types

(A) Breast, n = 22 vs. n = 981 TCGA-BRCA. (B) Genitourinary, n = 14 vs. n = 1567 TCGA-BLCA+KIRC+KIRP+ACC+PRAD. (C) Bladder urothelial, n = 5 vs. n = 410 TCGA-BLCA. (D) Adrenal, n = 4 vs. n = 88 TCGA-ACC. (E) Head and neck, n = 13 vs. n = 951 TCGA-HNSC+THCA. (F) Melanoma, n = 8 vs. n = 546 TCGA-SKCM+UVM. (G) Central nervous system (CNS) n = 5 vs. n = 896 TCGA-GBM+LGG. (H) Sarcoma, n = 7 vs. n=237 TCGA-SARC. Data is expressed as percentages values of samples affected with detected variants. Black bars represent mutation frequencies resulted from the TsT26 sequencing whereas white bars correspond to values obtained from the TCGA dataset.

Click here for additional data file.

Supplemental Information 6 Gene description of the TsT26 targeted panel and related genetic alterations identified in various tumor types

Click here for additional data file.

Supplemental Information 7 Set of 48 colorectal samples used in the inter-laboratory validation procedure

Click here for additional data file.

Supplemental Information 8 Tumor types dissected by histological types included in the set of 399 patients

Click here for additional data file.

Supplemental Information 9 Set of 399 samples tested with the Trusight Tumor26 during implementation

Click here for additional data file.

Supplemental Information 10 Clinical classification of detected variants

Click here for additional data file.

Supplemental Information 11 Clinical utility of the detected variants resulted from the TsT26 sequencing

Click here for additional data file.

Supplemental Information 12 Subsequent clinical decision-making to NGS-based panel results

Click here for additional data file.

Supplemental Information 13 Raw data of detected variants extracted from Illumina Variant studio software

Click here for additional data file.

We particularly would like to thank the participating patients for their collaboration in the present study. We are also grateful to pathology residents Ana Martínez López, Emilce Aguirre, Alvaro Pezella Risueño and María Trujillo Coronado for their assistance in variant classification during their rotation in the Fundación Jiménez Díaz University Hospital molecular diagnostics laboratory.

Additional Information and Declarations

Competing Interests

Author Contributions

Human Ethics

Data Availability

Víctor Manuel Fernández-Soria and Ricardo Ramos-Ruiz are employed by Genomics Unit, Madrid Science Park. The authors declare that they have no other competing interests.

Iván Prieto-Potin conceived and designed the experiments, analyzed the data, prepared figures and/or tables, authored or reviewed drafts of the paper, and approved the final draft.

Nerea Carvajal performed the experiments, analyzed the data, prepared figures and/or tables, and approved the final draft.

Jenifer Plaza-Sánchez, Carmen Laura Aúz-Alexandre, Laura Camacho, Raquel Longarón, Rosa Somoza and Víctor Manuel Fernández-Soria performed the experiments, prepared figures and/or tables, and approved the final draft.

Rebeca Manso performed the experiments, analyzed the data, prepared figures and/or tables, authored or reviewed drafts of the paper, and approved the final draft.

Cristina Chamizo, Sandra Zazo, Almudena López-Sánchez, Beatriz Bellosillo, Javier Hernández-Losa and Ricardo Ramos-Ruiz analyzed the data, prepared figures and/or tables, and approved the final draft.

Socorro María Rodríguez-Pinilla analyzed the data, authored or reviewed drafts of the paper, and approved the final draft.

Ion Cristóbal and Jesús García-Foncillas conceived and designed the experiments, authored or reviewed drafts of the paper, and approved the final draft.

Federico Rojo conceived and designed the experiments, analyzed the data, authored or reviewed drafts of the paper, and approved the final draft.

The following information was supplied relating to ethical approvals (i.e., approving body and any reference numbers):

Samples were obtained from the Biobank Fundación Jiménez Díaz (PT17/0015/0006), that belongs to the Spanish National Biobanks Network. The Fundación Jiménez Díaz University hospital Scientific Committee granted approval to carry out the study within its facilities, ethical application ref, PIC 23-2012.

The following information was supplied regarding data availability:

Sequencing raw data of detected variants extracted from Illumina Variant Studio software are available in Table S8.

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
