# Peer review of "Validation and clinical application of a targeted next-generation sequencing gene panel for solid and hematologic malignancies"

_PeerJ, doi:10.7717/peerj.10069_

## Round 0.1 · original submission · Major Revisions

Please ensure you respond to reviewer comments and especially the clinical utility of the test with such a high failure rate.

·

Basic reporting

Use of English throughout the manuscript needs improvement by the authors. The manuscript as submitted is difficult to understand in parts
e.g. Abstract section - ’The more repeated variant across GI and lung tumors were KRAS G12D/V/C.’
And
Line 19 - A crescent number of biomarkers are progressively required to characterize the molecular profile of a specific type of tumor or to administer targeted therapies (Morganti et al., 2018)
I recommend that the authors ask a native English speaker to review and re-phrase where required throughout the manuscript

Variants where described should follow HGVS guidelines for nomenclature throughout the manuscript. HGVS nomenclature is the accepted standard for variant identification - – see https://varnomen.hgvs.org/recommendations/DNA/.

I do not understand why read depth is expressed as kbp, this is not a standard measure of read depth in NGS sequence files where depth is expressed as 'N'x

The reference to Williams HL et al 2018 on line 315 is cited incorrectly using multiple author names

Experimental design

The paper is titled as a validation of a methodology. Clinical validations should include a test design specification with estimates of test sensitivity, specificity and where appropriate limit of detection arising from the validation and whether these met the design specification. There is no consideration of this in the manuscript. I feel it is essential to include these aspects in the manuscript.

The M&M section (Lines 54 – 179) are more detailed than I feel is needed given that the primary methods used are primarily manufacturer's standard protocols. I question whether this section needs to be so detailed.

Validity of the findings

I cannot agree with the authors conclusions in the Abstract (see quotation below) as the assay failure rate is unacceptably high.

‘Together, an appropriate validation of the TsT26 panel has granted a good application into the clinical routine by providing several relevant and potentially targetable variants across multiple FFPE tumors.’

The assay as described in the manuscript has a high sample inadequacy/failure rate of 1/3. In my opinion a 30% failure/sample inadequacy rate is incompatible with an assay for clinical application.

No explanation of the discordant results from the 48 reference samples tested across all 3 sites is given, the fact that the kit gives discordant results between the gold standard test and the TST26 for several samples e.g. 1-5, 3-4 & 3-11. The discordant results should be discussed and explained. From the data given a low VAF may explain the failure to detect KRAS variants with the gold standard method in 3-4 and 3-11 but no explanation of the differences between the two different methods is given which I feel is essential.

Reviewer 2 ·

Basic reporting

no comment

Experimental design

no comment

Validity of the findings

no comment

Additional comments

I carefully read the manuscript "Validation and clinical application of a targeted next generation
sequencing gene panel for solid and hematological malignancies".
This work is focused on demonstrating the applicability of the TruSight26 panel in the clinical setting. Although using the panel in this context requires several quality checks, the Authors were able to sequence two thirds of the samples subjected to the analytical workflow detecting 372 variants, 80% of them were tightly linked to potential FDA-approved clinical actionability, such as BRAF mutations V600 in melanoma and EGFR tyrosine-kinase domain variants in NSCLC.

The manuscript is clearly written, and the literature referenced is relevant and updated. Methods are described with sufficient detail and information to replicate them.
The topic addressed in the paper is of great interest to readers, and although several authors published on the applicability of the TruSight26 panel in the clinical and diagnostic field (Swati Garg et al., 2020. J Mol Diagn. 2020; Fisher KE et al., 2016. Journal of Molecular Diagnostics 18:299-315), the paper gives an important and useful contribution.

I believe this manuscript may represent a useful study for the future diagnostic application of the TruSight26 panel in diagnostic, which deserves publication after a revision.

Minor comments

• The English language and punctuation should be improved to ensure a clearer understanding of the text. Some examples:lane 309-311 ” Although the TsT26 panel lacks from the study of the gene TERT, BRAF and NRAS were the more frequently mutated genes in melanoma samples, as supported by others”; lane 324-325 “Indeed, the more validated NGS studies following its application in the everyday practice, the better the way to integrate the NGS technology into the clinics”.. the current phrasing makes comprehension difficult.
• In the Abstract, Methods section is reported “tumors types”or tumor types?
• Page 5, lane 13. Statement “On the other hand, the cost-effectivity..”: cost-effectivity or cost-effectiveness? Please, correct it.
• Among the supplementary files there is only the data S2 file. The data S1 file is not available.
• Page 10, lane 125. Statement “300x or greater for a single amplicon”: “At least 400 quality reads are necessary to detect a heterozygous variant with 99.9% sensitivity in a cancer specimen containing 10% neoplastic nuclei (Lin M.Tet al., 2014. Clinical validation of KRAS, BRAF, and EGFR mutation detection using next-generation sequencing. Am J Clin Pathol. 2014;141:856–866)”. Authors should kindly add a comment on this different coverage threshold.
• Materials and Methods section: Several articles of the Material and Methods section are not listed in the References section. Please check it.
• Page 13, lane 185-186. Statement “Sample 1-5 variant G12A was not detected and samples 3-4 and 3-11 were identified as variant detected, respectively as G12S and G12V…”. How do you justify the differences between the centers with respect to samples 1-5, 3-4 and 3-11? Authors introduce a comment, please?
• Page 14, lane 214-216. Statement “Thirty-seven samples did not undergo DNA quality assessment, 5% of them resulting in a NGS fail, 3% in detected variants and 1% in not detected variant (Table 4)”. Samples that failed the quality controls and were not sequenced or were sequenced with poor results had particular characteristics? In the work of Kevin E. Fisher et al., 2016 has been reported that a delta qCT value < or = to 6.0 is a practical adequacy requirement for TST library preparation and that the most of the specimens that failed library preparation were fine-needle aspiration samples, biopsy specimens, or body fluid cytospin cell blocks, then tissue and/or nonuniform sample. I would suggest the Authors to add a comment on the samples that have not generated sequenceable libraries.
• Page 15, lane 227-228. Statement “Thirty-seven runs were done during the clinical implementation, 17 employing v2 300-cycle and 20 using v3 600-cycle sequencing chemistries”. The Authors should comment briefly on whether or not they found differences in sequencing quality using the two chemistries.
• Page 15, lane 238-241. Statement “The mean coverage of AKT1 exon 2, STK11 exons 1 and 6 did not satisfied the minimum coverage of 1000x required by the panel. In addition…”. How did the authors decide to analyze the exons of the genes that did not satisfy the minimum coverage of 1000X?
• Figure S3: It is necessary to add a legend to figure S3 in order to make it legible and to better understand its meaning.

---

## Round 0.2 · Minor Revisions

Please revise as per reviewer 1 mainly for use of English.

·

Basic reporting

This is a re-review of a manuscript which on first review amongst other things, I felt needed improvement in the use of English by the authors who I accept are non-native English speakers. Although the authors have made some improvements to phrasing of points I highlighted in my first review there continues to be tracts of text that are difficult to follow. I understand that there is a PeerJ service for English proof-reading and editing, I recommend that the authors use this service to improve the readability of the whole manuscript.

I have highlighted several English and typographical errors below to illustrate this point however there are too many to list all of these here

Lines 228/9
Regarding the varied tumor types, the most abundant consisted in gastrointestinal (GI)
I suggest should read
Regarding the varied tumor types, the most abundant consisted of gastrointestinal (GI)

Line 231
cancer types were limited in number (Table S3). Overall consisted in biopsy specimens, surgical
I suggest should read
cancer types were limited in number (Table S3). Overall the sample set consisted of biopsy specimens, surgical

Line 251
whereas not detected variants or a wild type genotype was found in 26% (69 samples)
I suggest should read
whereas in 26% (69 samples) variants were either not detected or a wild type genotype was found

Line 252
The major number of detected variants was observed in TP53 (28%), KRAS (16%),
I suggest should read
The highest number of detected variants was observed in TP53 (28%), KRAS (16%),

Experimental design

The authors have taken on board my comments from the original review and included further details of the assay sensitivity and specificity. The revised manuscript benefits from this.

I am now happy with the experimental design

Validity of the findings

The authors have taken on board critical suggestions from my first review on the clinical utility of the assay considering the high failure rate of teh assay the authors have also commented and provided more explanation of genotype discordances between the TsT26 assay and the reference assays for KRAS variants.

I am now satisfied with the validity of the findings

Reviewer 2 ·

Basic reporting

No comment

Experimental design

NO comment

Validity of the findings

NO comment

Additional comments

The authors responded adequately to the referees' requests, in my opinion the manuscript can be accepted in the current form.

---

## Round 0.3 · accepted · Accept

Your article was improved from reviewer comments.